# Effects of Repetitive Transcranial Magnetic Stimulation on Pallidum GABAergic Neurons and Motor Function in Rat Models of Kernicterus

**DOI:** 10.3390/brainsci13091252

**Published:** 2023-08-28

**Authors:** Nanqin Wang, Yongzhu Jia, Xuanzi Zhou, Xia Wang, Huyao Zhou, Nong Xiao

**Affiliations:** Department of Rehabilitation, Children’s Hospital of Chongqing Medical University, National Clinical Research Center for Child Health and Disorders, Ministry of Education Key Laboratory of Child Development and Disorders, China International Science and Technology Cooperation Base of Child Development and Critical Disorders, Chongqing Key Laboratory of Translational Medical Research in Cognitive Development and Learning and Memory Disorders, Chongqing 400010, China; w13452339625@163.com (N.W.); jiayongzhu202306@163.com (Y.J.); xuanzi_zhou@163.com (X.Z.); connietwinkle@163.com (X.W.); zhou_huyao@163.com (H.Z.)

**Keywords:** kernicterus, rTMS, motor function, GAD67, apoptosis

## Abstract

Kernicterus is a serious complication of hyperbilirubinemia, caused by neuronal injury due to excessive unconjugated bilirubin (UCB) in specific brain areas. This injury induced by this accumulation in the globus pallidus can induce severe motor dysfunction. Repetitive transcranial magnetic stimulation (rTMS) has shown neuroprotective effects in various neurological diseases. This study aimed to investigate the effects of rTMS on pallidal nerve damage and motor dysfunction in a rat model of kernicterus. Rats were divided into a sham group (n = 16), a model group (bilirubin with sham rTMS; n = 16) and an rTMS group (bilirubin with rTMS; n = 16). High-frequency rTMS (10 Hz) was applied starting from 24 h postmodeling for 7 days. The rotarod test, western blotting and immunohistochemical staining were performed to measure motor function and protein expression levels. The rTMS mitigated the negative effects of UCB on the general health of kernicterus-model rats and improved their growth and development. Furthermore, the rTMS alleviated UCB-induced motor dysfunction and increased the expression of GABAergic neuronal marker GAD67 in the globus pallidus. Notably, it also inhibited apoptosis-related protein caspase-3 activation. In conclusion, rTMS could alleviate motor dysfunction by inhibiting apoptosis and increasing globus pallidus GAD67 in kernicterus rat models, indicating that it may be a promising treatment for kernicterus.

## 1. Introduction

Hyperbilirubinemia is a common and mostly benign condition, manifesting with yellow discoloration caused by bilirubin deposition in the skin, the sclera and other organs throughout the body, and it occurs in approximately 60% of term and 80% of preterm infants [1]. Conditions that increase bilirubin production or impede its elimination, like polycythemia, hemolysis caused by Rh alloimmunization or deficiencies in bilirubin metabolism enzymes, can induce hyperbilirubinemia [2,3]. During severe neonatal hyperbilirubinemia, unconjugated bilirubin (UCB) crosses the blood–brain barrier to penetrate and damage specific brain regions, resulting in irreversible brain injury, termed kernicterus [4]. Due to its lipid solubility, UCB is capable of crossing the immature blood–brain barrier and entering the brain tissue. Excessive UCB induces oxidative stress damage, impacting mitochondrial oxidative metabolism, suppressing cellular oxygen utilization, leading to cellular energy depletion and triggering intracellular calcium overload, which is the main cause of neuron damage [3]. Phototherapy stands as a widely accepted and effective primary treatment for neonatal jaundice [5]. Although phototherapy and exchange transfusion have therapeutic effects, kernicterus still causes disability and even death [6]. It remains a persistent issue in the contemporary world. It not only persists in developing countries with underdeveloped healthcare systems and health organizations disrupted by the impacts of war but also maintains its prevalence in industrialized nations [7]. In some developing nations, the incidence of severe kernicterus is roughly 100 times as high as it is in the developed world [3]. Severe neurological sequelae have serious impacts on patients’ quality of life and impose heavy burdens on society and families.

Kernicterus presents as abnormal motor movements and muscle tone, auditory disorder, oculomotor impairments and deciduous tooth enamel dysplasia [8]. The corresponding dystonia is mainly caused by globus pallidus lesions in the basal ganglia [9]. Gamma-aminobutyric acid-ergic (GABAergic) neurons are the main neurons in the globus pallidus [10,11]. Due to its neurotoxic selectivity, UCB induces neuronal apoptotic death in the globus pallidus via the intrinsic mitochondrial pathway through caspase-3 activation [12,13]. Several studies have reported the efficacy of minocycline in preventing neurological impairment and death in mouse models of severe neonatal hyperbilirubinemia [14]. Nevertheless, the prophylactic use of minocycline for the treatment of hyperbilirubinemia to prevent acute bilirubin encephalopathy or kernicterus is improbable due to its adverse effects on dental and dermal health [15]. In addition, few therapeutic options exist for patients with kernicterus who suffer from moderate to severe motor dysfunction. The current therapies for these patients encompass oral medications, particularly benzodiazepines, such as diazepam or clonazepam, alongside baclofen, which activates GABA-B receptors. The application of selective botulinum toxin aims to enhance function and alleviate painful muscle hypertonia [16]. Thus, our study aimed to explore a novel potential therapeutic approach for mitigating UCB-induced neuronal demise in the globus pallidus.

Repetitive transcranial magnetic stimulation (rTMS) is a safe and non-invasive therapeutic technique. It utilizes magnetic pulses to induce an electrical field within the brain via electromagnetic induction, thereby eliciting or modulating neural activity. Furthermore, the magnetic fields generated by conventional rTMS devices can penetrate the cerebral cortex at a depth of up to 1–2 cm [17,18,19]. Previous research has reported that rTMS decreased the quantity of caspase-3 positive cells in a rat model of transient cerebral ischemia [20]. Recent findings have further indicated that rTMS exhibits a neuroprotective effect through the downregulation of proapoptotic caspase-3 cleavage, thereby diminishing both infarct volume and cell death in the identical model [21]. Therefore, the objective of this study was to investigate whether rTMS can protect GABAergic neurons by mitigating UCB-induced cell death in the globus pallidus, ultimately improving the motor ability of kernicterus-model rats.

## 2. Materials and Methods

### 2.1. Animals

Pregnant Sprague Dawley (SD) rats were procured from the Experimental Animal Center of Chongqing Medical University (license No. SCXK [Yu] 2022-0010) and subsequently housed in a specific pathogen-free (SPF) animal laboratory. The rats were maintained under a 12 h light/dark cycle at a temperature of 23 ± 2 °C, with ad libitum access to food and water until natural birth. These experimental procedures received approval from the Ethics Committee of the Children’s Hospital of Chongqing Medical University (CHCMU-IACUC20230529003). All efforts were undertaken to minimize both the number of animals utilized and their distress.

### 2.2. Kernicterus Model

Bilirubin (Sigma-Aldrich, St Louis, MO, USA) was dissolved in 0.5 M NaOH solution (100 mg/mL), further diluted to a concentration of 10 mg/mL with ddH_2_O and subsequently adjusted to a pH of 8.5 with 0.5 M HCl. The solution was then stored in darkness at −20 °C [22].

On postnatal day 5, the rat pups (weighing 9–12 g) were randomly allocated into three groups: the sham group (n = 16), the model group (bilirubin with sham rTMS; n = 16) and the rTMS group (bilirubin with rTMS, n = 16). The pups were anesthetized with diethyl ether before the model establishment. The rats in the model group and the rTMS group each received an intracisternal injection of a 10 μg/g (body weight) bilirubin solution into the cisterna magna using a microinjector. The sham group was injected with equal volumes of ddH_2_O (pH = 8.5). Before the bilirubin or ddH_2_O was applied, an equal volume of the cerebrospinal fluid specimen was drained to prevent intracranial hypertension [23]. The complete experimental process is shown in Figure 1.

### 2.3. rTMS

A magnetic stimulator (CCY-II; Yiruide, Wuhan, China) with a circular coil (Y064; Yiruide, Wuhan, China) was applied to administer stimulation to conscious rats 24 h after model establishment. The coil was placed horizontally above each head, in close proximity to the scalp. The rats in the rTMS group received a total of 960 stimuli per day for 7 consecutive days at an intensity of 13% of the maximum stimulator output, including 48 trains at 10 Hz for 2 s, with 5 s intervals between the trains. The rats in the model group and the sham group underwent a similar procedure as those in the rTMS group, whereby the coil was vertically positioned 30 cm above the head. This elicited comparable auditory and tactile sensations but did not generate a magnetic field. The rats showed no signs of discomfort during the rTMS. They were sacrificed at seven days after the bilirubin injection for western blotting and immunohistochemistry. At postnatal day 28, the rats were subjected to the rotarod test to detect their motor ability.

### 2.4. Western Blotting (WB)

According to the instructions, the total protein was extracted from the globus pallidus with a total protein extraction kit (KGP250; KeyGen Biotech, Nanjing, China), and a bicinchoninic assay (BCA) protein detection kit (KGPBCA; KeyGen Biotech, Nanjing, China) was used to quantify the protein concentrations. Target proteins were isolated using sodium dodecyl sulphate–polyacrylamide gel electrophoresis (SDS–PAGE) and subsequently transferred to a polyvinylidene difluoride (PVDF) membrane. After cleaning with Tris-buffered saline with a Tween 20 (TBST) solution, the PVDF membranes were sealed in a blocking solution for 15 min. They were incubated overnight at 4 °C with anti-GAD67 (1:1000; Abcam, Cambridge, UK), anti-caspase-3 (1:1000; Abcam, Cambridge, UK) and anti-β-actin (1:2000; ABclonal, Wuhan, China) antibodies. The next day, the PVDF membranes were washed with TBST and incubated with the respective secondary antibodies for 1 h. Subsequently, each PVDF membrane was treated with an enhanced chemiluminescence (ECL) solution and exposed using the Bio-Rad ChemiDoc Touch System (Bio-Rad Laboratories, Hercules, CA, USA). Image analysis was performed using ImageJ software (Version 1.53; NIH, Bethesda, MD, USA).

### 2.5. Immunohistochemistry

The rats were sacrificed and then perfused with phosphate-buffered saline (PBS) followed by 4% paraformaldehyde solution (PFA) 7 days after modeling. Whole brains were removed and fixed with 4% paraformaldehyde for 48 h after sacrifice. Subsequently, 4 μm-thick paraffin sections of the globus pallidus were cut following paraffin embedding, with reference to the rat brain atlas [24]. Immunohistochemistry assays were performed using the universal two-step detection kit (ZSGB-BIO, Beijing, China). To evaluate the immunopositivity for GAD67 in the globi pallidi, the paraffin sections designated for immunohistochemistry were subjected to 1 h of baking at 60 °C. After deparaffinization and rehydration, heat-mediated antigen retrieval was performed with a Tris/EDTA buffer at pH 9.0, and then the sections were incubated overnight at 4 °C with the anti-GAD67 (1:500; Abcam, Cambridge, UK) antibody. Subsequently, the sections were incubated with the HRP-conjugated secondary antibody at room temperature for 30 min. Following rinsing with phosphate-buffered saline (PBS), the sections were treated with diaminobenzidine (DAB substrate kit; Abcam, Cambridge, UK) for 10 min and counterstained with hematoxylin for 1 min 30 s. Immunopositivity was examined with a slide scanning image system (Teksqray, Shenzhen, China), and representative images were displayed. For a clearer presentation of the results, Image-Pro Plus 6.0 (Media Cybernetics; Washington, D.C., USA) was used to calculate the mean optical density of the positive areas.

### 2.6. Body Weight

The body weights of the rats were recorded daily from day 5 to day 13 after birth to assess their general health during the rTMS intervention.

### 2.7. Rotarod Test

The rotarod test was performed at 28 days of age to evaluate motor ability. The rats were allowed to acclimate to the environment and instrument 1 day before this experiment. In the formal test, the rats were placed on a rotarod, which gradually increased in speed from 10 to 80 rpm over 3 min. This test concluded if the rat fell off the rotating rod, completed a full revolution while clinging to the rod or successfully remained on the rod for the entire 3 min. The equipment was cleaned with 70% ethanol before and after the testing of each rat. The duration of time that each rat stayed on the rotating rod was recorded, and this experiment was repeated three times to calculate the mean duration.

### 2.8. Statistical Analysis

Statistical analysis and graphing were performed using GraphPad Prism 9.0 (GraphPad Software, La Jolla, CA, USA). The data were expressed as means ± standard deviations (SDs). Group comparisons were assessed using one-way ANOVA followed by Tukey’s test for multiple comparisons. For the body weight data, two-way repeated-measures ANOVA, followed by Tukey’s multiple comparison test, was employed. Statistical significance was defined as *p* < 0.05.

## 3. Results

### 3.1. rTMS Improved the General Health and Motor Ability of Rats with Kernicterus

On the first day after modeling, the body weights of the model group and the rTMS group significantly decreased compared to the sham group (*p* < 0.001). Subsequently, although the body weights of the rats in the model group and the rTMS group gradually increased, they remained lower than those in the sham group (*p* < 0.001). The rTMS group, which underwent rTMS intervention, exhibited significant increases in body weight and faster recovery (*p* < 0.05) (Figure 2A). In the present experiment, the injection of UCB through the cisterna magna had a detrimental effect on the general health of the rats while the rTMS mitigated the damage caused by the UCB and promoted their recovery.

The ultimate goal of this treatment was to improve motor ability, which was assessed using the rotarod test. The results revealed a significant increase in the residence times of the rats in the rTMS group compared to the model group (*p* < 0.05). Furthermore, there were no significant differences in the time spent on the rotarod between the rTMS group and the sham group (Figure 2B). These results suggest that the injection of UCB through the cisterna magna resulted in impaired motor ability, which could be mitigated by rTMS.

### 3.2. rTMS Increased the Expression of GAD67 in the Globi Pallidi of Rats with Kernicterus 

The effects of the rTMS on the pallidal GABAergic neurons were assessed using western blotting and immunohistochemistry to detect the expression of GAD67 (GABAergic neuronal marker). The western blotting revealed a significant decrease in the GAD67 expression at 7 days after modeling (*p* < 0.001), while the rTMS treatment demonstrated a significant increase in the GAD67 expression (*p* < 0.05). Immunohistochemistry further supported these results by showing an increased number of GABAergic neurons in the globi pallidi in the rTMS group compared to the model group. The rTMS group showed stronger immunohistochemical staining for GAD67 in the globus pallidus than that in the model group. Quantitative analysis corroborated the western blotting results, confirming a significant upregulation of GAD67 in the rTMS group compared to the model group (*p* < 0.05) (Figure 3). The present results demonstrate that the UCB impaired the GABAergic neurons in the globus pallidus, while the rTMS were able to ameliorate the UCB-induced GABAergic neuronal lesions in the same area.

### 3.3. rTMS Inhibited the Activation of Caspase-3

To investigate the potential mechanisms underlying the protective effects of rTMS on GABAergic neurons in the globus pallidus, we assessed caspase-3, a protein associated with apoptotic cell death. At 7 days after modeling, the western blotting demonstrated a significant increase in cleaved Caspase-3 expression in the model group compared to the sham group (*p* < 0.001). Conversely, rats treated with rTMS exhibited downregulation of cleaved Caspase-3 expression compared to the model group (*p* < 0.05) (Figure 4).

## 4. Discussion

To our knowledge, this study is the first to have confirmed the protective effects of rTMS on pallidal GABAergic neurons in rats with kernicterus: improved general health and motor ability. The findings of this study revealed that the rTMS increased the protein expression levels of GAD67 while simultaneously inhibiting caspase-3 activation. The mechanism described in this study is shown in diagram form in Figure 5. In summary, rTMS could be a promising treatment for kernicterus, with the potential to inhibit apoptosis and exhibit protective effects on GABAergic neurons in the globus pallidus, thereby alleviating UCB-induced motor dysfunction.

Kernicterus is caused by the accumulation of excessive UCB in the brain during severe neonatal hyperbilirubinemia [8]. It manifests as a preference for bilirubin neurotoxicity toward neurons and exhibits a regional pattern of UCB-induced neuronal injury, prominently affecting the globus pallidus, subthalamic nucleus, hippocampus, oculomotor nuclei, ventral cochlear nuclei and cerebellum [25]. Clinical manifestations of kernicterus include dystonia, choreoathetosis, auditory processing disturbances (with or without hearing loss), oculomotor impairments and dysplasia of the enamel of deciduous teeth [26]. It is believed that kernicteric dystonia arises from neurotoxic damage caused by UCB in the globus pallidus. Human magnetic resonance imaging (MRI) and autopsies have revealed the presence of bilateral lesions in the globus pallidus [27,28,29,30]. Furthermore, there has been a suggestion that kernicteric dystonia results from decreased output of the globus pallidus, which subsequently affects the input to downstream motor nuclei, causing abnormal movements and muscle tension [31]. UCB-induced neurological dysfunction encompasses various factors, including oxidative stress, neuroexcitotoxicity, neuroinflammation, mitochondrial energy failure, etc. [32,33,34,35]. UCB increases and prolongs the presence of the excitatory neurotransmitter glutamate in the synaptic cleft. Excessive glutamate overstimulates neuronal N-methyl-D-aspartate (NMDA) receptors, leading to increased influxes of sodium, calcium, chloride and water. This process generates free radicals and activates caspase-3, resulting in apoptosis [36]. UCB also interacts with neuronal cell membranes, inducing oxidative damage to these membranes and increased permeability. This interaction compromises the integrity of membrane-associated lipoprotein structures, disrupts cellular homeostasis and ultimately triggers neuronal apoptosis [37]. This study aimed to investigate the hypothesis that rTMS could ameliorate the motor dysfunction induced by UCB and exert a neuroprotective effect on the GABAergic neurons in the globi pallidi of rats with kernicterus. Our findings demonstrated that rTMS mitigated UCB-induced motor impairments and led to improved general health in rats with kernicterus. Additionally, our histological investigation revealed a higher preservation of GABAergic neurons in the globi pallidi in the rTMS group compared to the model group, indicating the neuroprotective effect of rTMS. Moreover, we have presented evidence showing that the rTMS significantly attenuated caspase-3 activation in the globus pallidus. These data demonstrate that rTMS may reduce UCB-induced loss of GABAergic neurons in the globus pallidus, partly by regulating apoptotic pathways.

rTMS is a safe and non-invasive therapeutic technique that employs an electromagnetic field generated by coils to stimulate induced currents within the brain, causing ions to move across cell membranes. The induced current generated by rTMS causes membrane potential depolarization and hyperpolarization, thereby modulating neuronal activity in damaged areas of the brain and ultimately ameliorating neurological dysfunction [38,39,40]. In recent decades, rTMS has been extensively utilized in the study of neuropsychiatric disorders. The neuroprotective effect of rTMS involves the following mechanisms: the regulation of neurotransmitters, such as increased dopamine in various brain regions as well as glutamate and GABA in the striatum and hippocampus, respectively [41,42]; the modulation of gene expression, such as upregulating brain-derived neurotrophic factor (BDNF) gene expression [43,44]; the alleviation of oxidative stress [45,46]; and the reduction of neuroinflammation and apoptosis [47,48]. rTMS was reported to inhibit the neurotoxic transformations of astrocytes, leading to reduced neuronal apoptosis and infarct volumes in ischemic rats [49]. A previous study has shown that rTMS attenuated oxidative stress, causing decreased cell loss in the striatum in a 3-nitropropionic model of Huntington’s disease [47]. Animal studies of Parkinson’s disease have revealed that rTMS decreased the levels of cyclooxygenase-2 and tumor necrosis factor-α in rat substantia nigra [48], improved motor functions and preserved the function of dopamine neurons [50]. Additionally, rTMS has been found to reduce glutamate-driven excitotoxicity and motor neuron death in amyotrophic lateral sclerosis (ALS) [51]. The neuronal damage caused by UCB is known to be associated with oxidative stress, neuroexcitotoxicity and neuroinflammation in kernicterus. In this study, the neuroprotective effects of rTMS on pallidal GABAergic neurons may involve oxidative stress, neuroexcitotoxicity and neuroinflammation.

Motor dysfunction is one of the primary reasons for physical disability in kernicterus. In this study, we evaluated the motor ability of the rats by the rotarod test. It was reported that the kernicterus-model rats had significantly worse performances on the rotarod test than the controls, indicating that the kernicterus-model rats experienced motor function deficits [52]. The UCB-treated rats showed significant motor dysfunction in this test. After rTMS intervention, we found that the rats in the rTMS group performed better in the rotarod test than those in the model group, which indicates that the rTMS alleviated the UCB-induced motor dysfunction. A previous study has revealed that the body weights of rats with kernicterus were lower than those of a sham group [23], which is consistent with our results. Our data have shown that rTMS can reduce UCB-induced adverse effects on kernicterus-model rats’ body weights, suggesting that it could improve the growth and development of rats with kernicterus.

Irreversible damage becomes apparent in brain tissue 24 h after bilirubin injection, and the symptoms will become noticeably pronounced after seven days of modeling [53]. In this study, rTMS was conducted during the acute phase. Western blotting and immunohistochemistry were employed to evaluate the expression levels of GAD67 and apoptosis-related protein caspase-3 after the rTMS interventions. Early exposure to bilirubin would result in long-term motor function abnormalities; thus, behavioral tests were performed to evaluate the effects of rTMS on the long-term motor function of the model rats. Our data demonstrated a higher expression of GABAergic neuronal marker GAD67 in the rTMS group compared to the model group. The results of our study have suggested that rTMS may not only improve motor functions but also exert neuroprotective effects on GABAergic neurons in the globi pallidi of rats with kernicterus. The neuroprotective effects of rTMS on GABAergic neurons may be attributed to reductions in neuroexcitotoxicity, the attenuation of oxidative stress and neuroinflammation; the upregulation of neurotrophic factors such as the brain-derived neurotrophic factor (BDNF); and the mitigation of astrogliosis [35,36,54]. The globus pallidus is a primary target of UCB-induced injury in a rat model of kernicterus, primarily consisting of GABAergic and cholinergic neurons, with the GABAergic neurons accounting for 95% of the population. The neurotoxic impacts of unconjugated bilirubin can result in the damage or death of pallidal neurons, ultimately resulting in irreversible motor function impairment [55,56]. GABA serves as the principal inhibitory neurotransmitter in the basal ganglia, and abnormalities in GABAergic circuitry within these ganglia have been implicated in various neurological disorders and degenerative diseases, such as Huntington’s disease, Parkinson’s disease and hyperkinesia [57,58,59]. In the present study, the improvement of motor ability may be associated with the increased expression of GABAergic neuronal marker GAD67. Deep brain stimulation (DBS) has been established as an effective treatment technology for movement disorders [60,61]. Moreover, DBS leads are usually placed in the globus pallidus interna in the treatment of dystonia, which partly explains the increased expression of the pallidal neuron marker GAD67 and the improvement in motor function in our study. Previous studies have revealed that bilirubin injection could trigger the activation of caspase-3, particularly in the globus pallidus [53,62]. Therefore, in this study, pallidal GABAergic neurons were selected as the subjects of investigation for examination of the effects of rTMS on these neurons and their subsequent impacts on motor function. We found that rTMS suppressed the activation of caspase-3, suggesting that the neuroprotective effect of rTMS may involve the inhibition of the apoptotic program mediated by caspase-3. Despite this progress, the precise underlying mechanism remains unclear, necessitating further research to elucidate the detailed neuroprotective mechanism of rTMS in kernicterus.

There were several limitations to our study. First, it was conducted as a preliminary investigation, wherein we provided initial evidence of the neuroprotective effect of rTMS in rats with kernicterus. Thus, further research and verification are required to elucidate the specific mechanism involved. Second, despite our use of the smallest available animal-specific coil, rTMS in rodents has limitations that may result in whole-brain stimulation. Third, the sample size was relatively small; thus, our findings should be confirmed in future large-scale studies. Fourth, we omitted the inclusion of a sham + rTMS group, since GAD67 regulation is among the therapeutic effects of rTMS in rat models of kernicterus, which were our primary focus. The effects of rTMS in isolation have yet to be elucidated and warrant further investigation. Finally, we relied solely on the rotarod test to evaluate the motor ability of the rats. Considering the limitations of our study, further investigation is needed.

## 5. Conclusions

The present study has provided evidence that rTMS could alleviate motor dysfunction in rats with kernicterus. It may also exert neuroprotective effects on pallidum damage induced by UCB by inhibiting apoptosis and improving the function of the GABAergic neurons in the globus pallidus. Our findings suggest that rTMS is a promising treatment for kernicterus.

## Figures and Tables

**Figure 1 brainsci-13-01252-f001:**
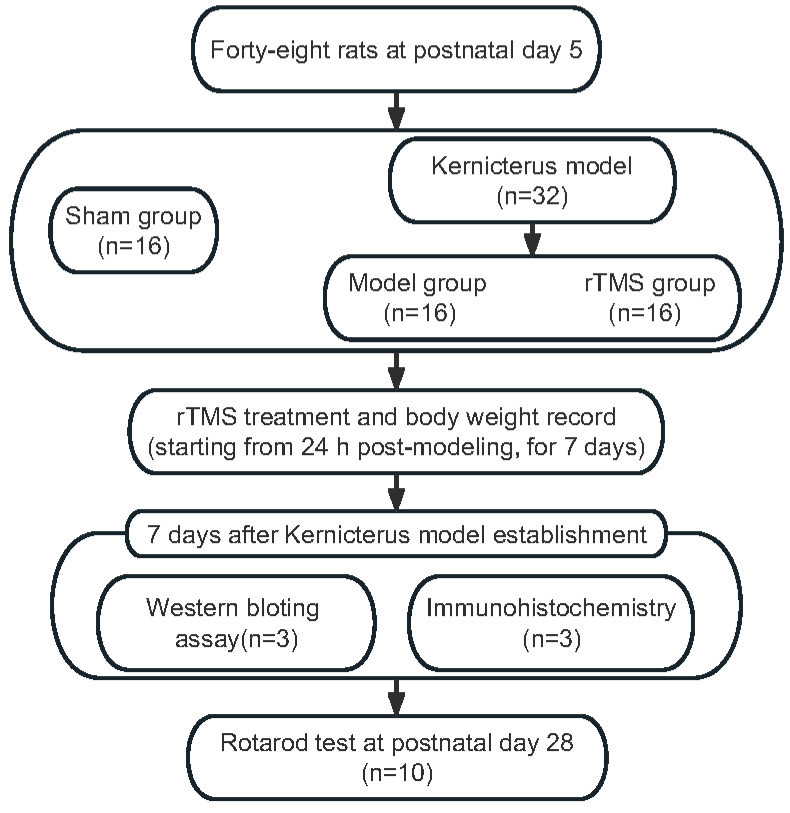
Experimental flow chart.

**Figure 2 brainsci-13-01252-f002:**
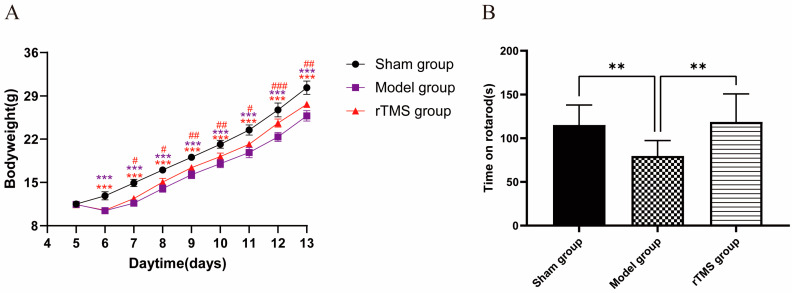
rTMS improves the general health and motor ability of rats with kernicterus. (**A**) The effects of rTMS on the body weights of rats with kernicterus. Data are expressed as means ± SDs; *** *p* < 0.001 vs. the sham group; ^#^
*p* < 0.05, ^##^
*p* < 0.01, ^###^
*p* < 0.001 vs. the model group; n = 5. (**B**) Time on rotarod in the rotarod test. Data are expressed as means ± SDs; ** *p* < 0.01; n = 10.

**Figure 3 brainsci-13-01252-f003:**
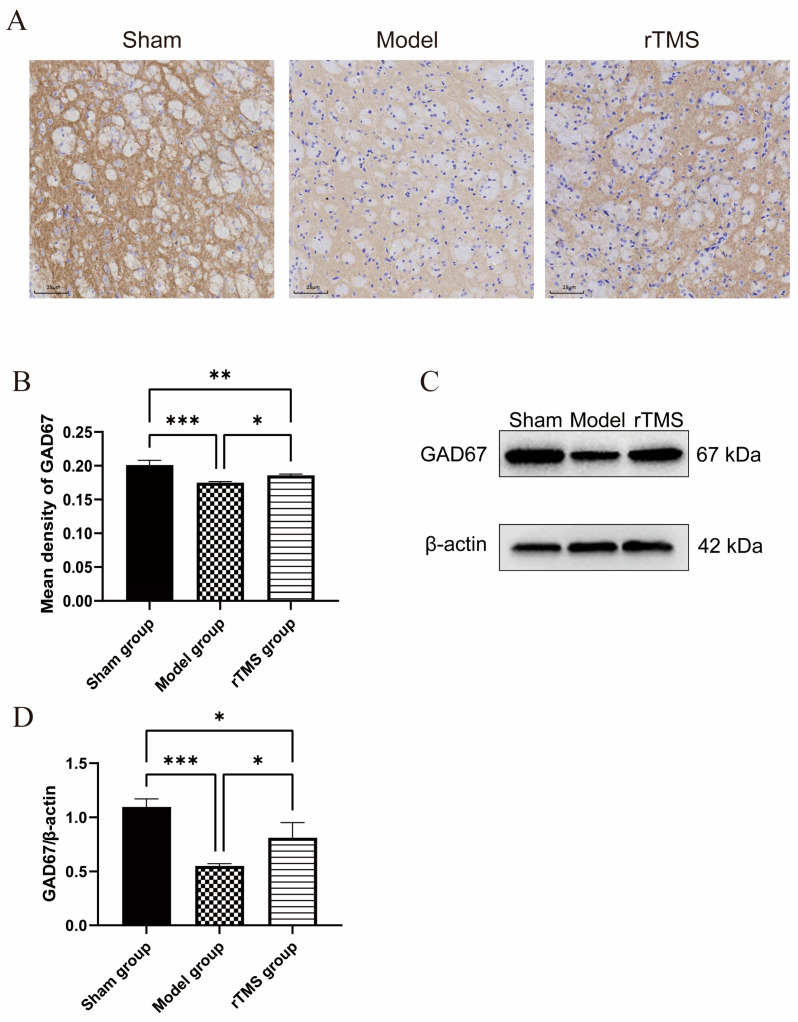
rTMS increases GABAergic neuronal marker GAD67’s expression level in the globi pallidi of rats with kernicterus. (**A**) Results of immunohistochemistry staining in the globus pallidus. Scale bars: 25 μm. (**B**) Quantitative analysis of GAD67 in the globi pallidi. Data are expressed as means ± SDs; n = 3. * *p* < 0.05, ** *p* < 0.01, *** *p* < 0.001. (**C**) Results of western blotting analysis of GAD67 expression in the globus pallidus. (**D**) Relative protein expression of GAD67, normalized with β-actin. Data are expressed as means ± SDs; n = 3. * *p* < 0.05, *** *p* < 0.001.

**Figure 4 brainsci-13-01252-f004:**
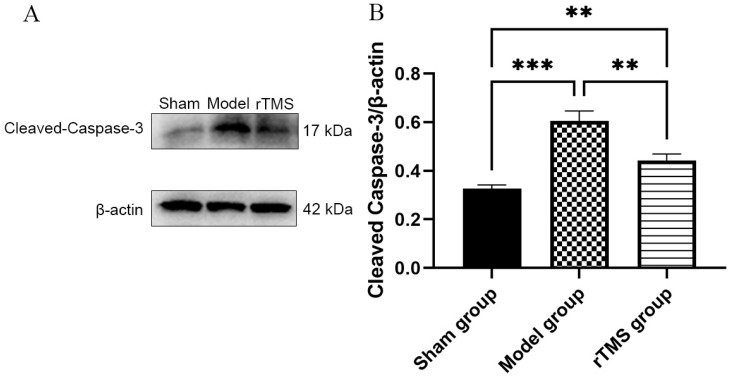
rTMS inhibits activation of caspase-3 in the globi pallidi of rats with kernicterus. (**A**) Results of western blotting analysis of cleaved Caspase-3 expression in the globus pallidus. (**B**) Relative protein expression of cleaved Caspase-3, normalized with β-actin. Data are expressed as means ± SDs; n = 3. ** *p* < 0.01, *** *p* < 0.001.

**Figure 5 brainsci-13-01252-f005:**
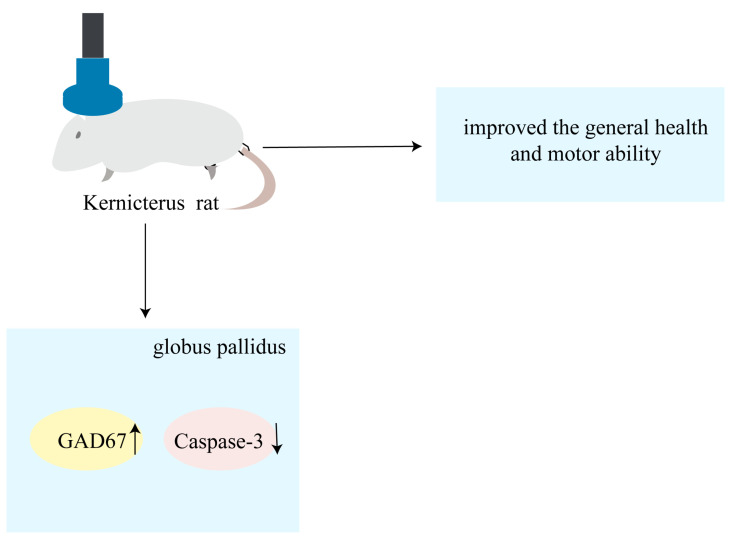
Schematic representation showing that rTMS alleviates motor dysfunction and improves general health by inhibiting apoptosis and increasing globus pallidus GAD67 in rat models of kernicterus.

## Data Availability

The data presented in this study are available from the corresponding authors upon reasonable request.

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
