# Peer review of "Effects of Repetitive Transcranial Magnetic Stimulation on Pallidum GABAergic Neurons and Motor Function in Rat Models of Kernicterus"

_brainsci, 2023, doi:10.3390/brainsci13091252_

Round 1

Reviewer 1 Report

In the manuscript submitted for review, the Authors describe "the effects of repetitive transcranial magnetic stimulation on the pallidum GABAergic neurons and motor function in rat models of kernicterus". The manuscript is interesting, written concisely and understandably, but I have the impression that the Authors described the results of their experiment very briefly.

My comments:

1. methodology: how the antigen was retrievel [line 129]

2. methodology: how long was hematoxylin stained [line 133]

3. the Authors took pictures, but there is no exact description of what they show, what are the differences between the groups, where is the positive reaction....and the scale in the pictures is illegible.

Author Response

Dear reviewer:

Thank you for reviewing our manuscript and for the constructive comments, which greatly helped us to improve the manuscript. Those comments are all valuable and very helpful for revising and improving our paper, as well as the important guiding significance to our researches. We have studied comments carefully and have made correction which we hope meet with approval. Revised portion are marked in red in the paper. The main corrections in the paper and the responds to the comments are as flowing:

 Point 1: methodology: how the antigen was retrievel [line 129]

Response 1: We used heat mediated antigen retrieval with Tris/EDTA buffer pH 9.0 before commencing with immunohistochemistry. [line 143]

Point 2: methodology: how long was hematoxylin stained [line 133]

Response 2: The sections were counterstained with haematoxylin for 1 min 30 s. [line 149]

Point 3: the Authors took pictures, but there is no exact description of what they show, what are the differences between the groups, where is the positive reaction....and the scale in the pictures is illegible.

Response 3: We added the description of the pictures. [line 210]

In oder to show the specific differences between the groups, quantitative analysis was used. [line 202]

The positive reaction is the brown staining area in the pictures.

We are very sorry for our negligence of illegible scale in the pictures. The scale is illegible due to picture compression, so we showed the specific scale bars in the legend notes. [line 211]

Special thanks to you for your good comments.

Reviewer 2 Report

Comments for authors

Brain Sci. – Manuscript ID: brainsci-2492218– “Effects of Repetitive Transcranial Magnetic Stimulation on the Pallidum GABAergic Neurons and Motor Function in Rat Models of Kernicterus.”, by Nanqin Wang, Yongzhu Jia, Xuanzi Zhou, Xia Wang, Huyao Zhou and Nong Xiao.

In this manuscript, the authors investigate whether repetitive transcranial magnetic stimulation (rTMS) can protect GABAergic neurons by mitigating unconjugated bilirubin (UCB) - induced cell death in the globus pallidus, ultimately improving the motor ability of kernicterus model rats. The present study provided evidence that rTMS could alleviate motor dysfunction of kernicterus rats. Furthermore, rTMS may exert neuroprotective effects on pallidum damage induced by UCB through inhibiting apoptosis and improving the function of GABAergic neurons in the globus pallidus. Auhtors suggest that rTMS is a promising treatment for kernicterus.

This is an interesting paper addressing an interesting perspective candidate for the efficient therapy in kernicterus and more generally in neurodegenerative diseases.

Methods are more or less relevant. They aren’t major flaws or biases and conclusions are based on the data. The literature is up-to-date. Then, the topic is suited to Brain Sci.

Authors report precisely author contributions, ethics board approval, disclosure of funding and conflicts of interest. There isn’t reason to suspect research misconduct.

Discussion and conclusions are critical and concise. Figures are explicit and they add to the message. Presentation logical and language are adequate.

Finally, the manuscript is well written and well presented, however, there are few issues that the authors should take into considerations and finally, some aspects need to be discussed. Below are some specific comments:

My main criticisms of this paper concern the method.

-          The control rats (sham group) is not really clear. In fact, did the animals in the control group receive the rTMS treatment? If not, in my opinion this group is missing. How to eliminate some effects of the rTMS alone, an induced hyperactivity or a GAD67 overexpression in all animals regardless of their "pathological” status?

-          Why don't all animals have the rotarod test at different stages of neuronal death progression and also why don't they all have post-mortem analysis?

In the study’s flow chart, it would be better to perform rotarod test 7 days after Kernicterus model establishment and also at postnatal day 28. Immunohistochemistry and western bloting should be performed on more animals than 3 at the end of the animal observation period.

-          Different analysis (behavioral, western bloting, immunohistochemistry) are performed at different time points during disease’s progression. So how is it possible to discriminate an effect related to the disease’s progression from an effect related to the treatment?

At a minimum, the authors must discuss the delay between the different time point analyzes and present the kinetics of neuronal death’s progression induced by the injection of bilirubin. Then they could conclude to rTMS neuroprotective effects on pallidum damage induced by UCB.

-          Moreover, by adding a sham+rTMS group, the authors will use a cleaner statistical test, a two-way ANOVA (with group and treatment factors) followed by the Tukey test for multiple comparisons. 

It seems to me absolutely necessary to take thess methodological points into account, at least to discuss them before considering publishing these results.

Author Response

Dear reviewer:

Thank you for reviewing our manuscript and for the constructive comments, which greatly helped us to improve the manuscript. Those comments are all valuable and very helpful for revising and improving our paper, as well as the important guiding significance to our researches. We have studied comments carefully and have made correction which we hope meet with approval. Revised portion are marked in red in the paper. The main corrections in the paper and the responds to the comments are as flowing:

 Point 1: The control rats (sham group) is not really clear. In fact, did the animals in the control group receive the rTMS treatment? If not, in my opinion this group is missing. How to eliminate some effects of the rTMS alone, an induced hyperactivity or a GAD67 overexpression in all animals regardless of their "pathological” status?

Response 1: The sham group received sham rTMS. In Discussion (line 252), Previous research found that the neuroprotective effect of rTMS involved the regulation of neurotransmitters including regulating GABA. GAD67 As GABAergic neuronal marker, GAD67 is also the key rate‐limitation enzyme in the process of GABA production. Since the regulation of GABA was one of the therapeutic effects of rTMS, we did not set up additional sham+rTMS group in order to minimize both the number of animals utilized and their distress. We added this part in the Discussion(line 279). Special thanks to you for your good comments.

Point 2: Why don't all animals have the rotarod test at different stages of neuronal death progression and also why don't they all have post-mortem analysis?

In the study’s flow chart, it would be better to perform rotarod test 7 days after Kernicterus model establishment and also at postnatal day 28. Immunohistochemistry and western bloting should be performed on more animals than 3 at the end of the animal observation period.

Response 2: Our emphasis was on investigating the effects of rTMS during the acute phase on the Kernicterus model rats, so we conducted rTMS during the acute phase, so we only assessed the expression of the target protein during this phase. The rotarod test was conducted to examine the effects of rTMS on long-term motor function. In addition, rats at 7 days post-modeling were not mature enough for the rotarod experiment. The rotarod test was typically conducted on postnatal day 28.

Point 3: Different analysis (behavioral, western bloting, immunohistochemistry) are performed at different time points during disease’s progression. So how is it possible to discriminate an effect related to the disease’s progression from an effect related to the treatment?

At a minimum, the authors must discuss the delay between the different time point analyzes and present the kinetics of neuronal death’s progression induced by the injection of bilirubin. Then they could conclude to rTMS neuroprotective effects on pallidum damage induced by UCB.

Response 3: Thanks to you for your good comments. Western blotting and immunohistochemistry were to analyze the effects of rTMS on the expression of the target protein during the acute phase. Behavioral test was conducted to evaluate the effect of rTMS on the long-term motor function of the model rats. According to the reviewer's comment, we added discussion.(line 349)

Point 4: Moreover, by adding a sham+rTMS group, the authors will use a cleaner statistical test, a two-way ANOVA (with group and treatment factors) followed by the Tukey test for multiple comparisons.

 Response 4: We appreciate your professional feedback on our manuscript. We mainly focus on the effect of rTMS on the regulation of GAD67 of Kernicterus model rats, so we did not set up sham+rTMS group. Thank you for your thorough and in-depth review of this manuscript. Your professional feedback and constructive suggestions have contributed to enhancing the quality of the paper. Your suggestions are highly valuable for our current and future research endeavors.

Special thanks to you for your good comments.

Reviewer 3 Report

The original article by Wang et al. "Effects of Repetitive Transcranial Magnetic Stimulation on the Pallidum GABAergic Neurons and Motor Function in Rat Models of Kernicterus" covers a potentially interesting and emerging topic related to the kernicterus therapy. In this sense, this remains to be potentially interesting for the Brain Sciences readers. I regard the main point of this paper as highly attractive as well as the results are clearly presented. The text does not contain any major errors, therefore I have some minor comments and recommendations:

1. There is a need to provide slightly more expanded introduction shortly
mentioning/describing pathogenesis of kernicterus and its impact of modern healthcare.

2. The figure summarizing and clarifying the results should be added.

3. Following references should be added and properly cited within the main text to improve the quality of manuscript

- Mela A, Poniatowski ŁA, Drop B, Furtak-Niczyporuk M, Jaroszyński J, Wrona W, Staniszewska A, Dąbrowski J, Czajka A, Jagielska B, Wojciechowska M, Niewada M. Overview and Analysis of the Cost of Drug Programs in Poland: Public Payer Expenditures and Coverage of Cancer and Non-Neoplastic Diseases Related Drug Therapies from 2015-2018 Years. Front Pharmacol. 2020 Aug 14;11:1123. doi: 10.3389/fphar.2020.01123.

- Kasirer Y, Kaplan M, Hammerman C. Kernicterus on the Spectrum. Neoreviews. 2023 Jun 1;24(6):e329-e342. doi: 10.1542/neo.24-6-e329.

- Mela A, Rdzanek E, Poniatowski ŁA, Jaroszyński J, Furtak-Niczyporuk M, Gałązka-Sobotka M, Olejniczak D, Niewada M, Staniszewska A. Economic Costs of Cardiovascular Diseases in Poland Estimates for 2015-2017 Years. Front Pharmacol. 2020 Sep 8;11:1231. doi: 10.3389/fphar.2020.01231.

-Gottimukkala SB, Lobo L, Gautham KS, Bolisetty S, Fiander M, Schindler T. Intermittent phototherapy versus continuous phototherapy for neonatal jaundice. Cochrane Database Syst Rev. 2023 Mar 2;3(3):CD008168. doi: 10.1002/14651858.CD008168.pub2.

4. In some places the use of English throughout the whole manuscript could be improved on.

Completing this gaps will have an impact on the understanding the aim of the study and, from
my point of view, is absolutely necessary.

minor review

Author Response

Dear reviewer:

Thank you for reviewing our manuscript and for the constructive comments, which greatly helped us to improve the manuscript. Those comments are all valuable and very helpful for revising and improving our paper, as well as the important guiding significance to our researches. We have studied comments carefully and have made correction which we hope meet with approval. Revised portion are marked in red in the paper. The main corrections in the paper and the responds to the comments are as flowing:

 Point 1: There is a need to provide slightly more expanded introduction shortlymentioning/describing pathogenesis of kernicterus and its impact of modern healthcare.

Response 1: We expanded the introduction according to the reviewer’s comments.(line 41, 47)

Point 2: The figure summarizing and clarifying the results should be added.

Response 2: We appreciate your professional feedback on our manuscript. The figure summarizing and clarifying the results was added(Figure 5)(line 236).

Point 3: Following references should be added and properly cited within the main text to improve the quality of manuscript

- Mela A, Poniatowski ŁA, Drop B, Furtak-Niczyporuk M, Jaroszyński J, Wrona W, Staniszewska A, Dąbrowski J, Czajka A, Jagielska B, Wojciechowska M, Niewada M. Overview and Analysis of the Cost of Drug Programs in Poland: Public Payer Expenditures and Coverage of Cancer and Non-Neoplastic Diseases Related Drug Therapies from 2015-2018 Years. Front Pharmacol. 2020 Aug 14;11:1123. doi: 10.3389/fphar.2020.01123.

- Kasirer Y, Kaplan M, Hammerman C. Kernicterus on the Spectrum. Neoreviews. 2023 Jun 1;24(6):e329-e342. doi: 10.1542/neo.24-6-e329.

- Mela A, Rdzanek E, Poniatowski ŁA, Jaroszyński J, Furtak-Niczyporuk M, Gałązka-Sobotka M, Olejniczak D, Niewada M, Staniszewska A. Economic Costs of Cardiovascular Diseases in Poland Estimates for 2015-2017 Years. Front Pharmacol. 2020 Sep 8;11:1231. doi: 10.3389/fphar.2020.01231.

-Gottimukkala SB, Lobo L, Gautham KS, Bolisetty S, Fiander M, Schindler T. Intermittent phototherapy versus continuous phototherapy for neonatal jaundice. Cochrane Database Syst Rev. 2023 Mar 2;3(3):CD008168. doi: 10.1002/14651858.CD008168.pub2.

Response 3: Thank you for offering the references. We cited the references within the main text.(line 41, 45)

Point 4: In some places the use of English throughout the whole manuscript could be improved on.

 Response 4: Thanks to you for your good comments. We have modified our sentences in the revised manuscript and tried our best to improve the manuscript.

Special thanks to you for your good comments.

Round 2

Reviewer 2 Report

Comments for authors

Brain Sci. – Manuscript ID: brainsci-2492218-peer-review-v2– “Effects of Repetitive Transcranial Magnetic Stimulation on the Pallidum GABAergic Neurons and Motor Function in Rat Models of Kernicterus.”, by Nanqin Wang, Yongzhu Jia, Xuanzi Zhou, Xia Wang, Huyao Zhou and Nong Xiao.

In the first version of the manuscript, the authors report investigations on repetitive transcranial magnetic stimulation (rTMS) effects on GABAergic neurons. They report the protective effects of rTMS on pallidal GABAergic neurons in rats with kernicterus by mitigating unconjugated bilirubin (UCB) - induced cell death in the globus pallidus, ultimately improving the motor ability of kernicterus model rats. The present study provided evidence that rTMS could alleviate motor dysfunction of kernicterus rats. Furthermore, rTMS may exert neuroprotective effects on pallidum damage induced by UCB through inhibiting apoptosis and improving the function of GABAergic neurons in the globus pallidus. Auhtors suggest that rTMS is a promising treatment for kernicterus.

The authors submitted a carefully-prepared revision, which satisfactorily addressed the remaining concerns.

 They have addressed all comments.

Therefore, I recommend publication
